# The Potential of Deep Learning to Advance Clinical Applications of Computational Biomechanics

**DOI:** 10.3390/bioengineering10091066

**Published:** 2023-09-09

**Authors:** George A. Truskey

**Affiliations:** Department of Biomedical Engineering, Duke University, Durham, NC 27701, USA; gtruskey@duke.edu; Tel.: +1-919-660-5147

**Keywords:** biomechanics, finite element models, neural networks, deep learning, image segmentation

## Abstract

When combined with patient information provided by advanced imaging techniques, computational biomechanics can provide detailed patient-specific information about stresses and strains acting on tissues that can be useful in diagnosing and assessing treatments for diseases and injuries. This approach is most advanced in cardiovascular applications but can be applied to other tissues. The challenges for advancing computational biomechanics for real-time patient diagnostics and treatment include errors and missing information in the patient data, the large computational requirements for the numerical solutions to multiscale biomechanical equations, and the uncertainty over boundary conditions and constitutive relations. This review summarizes current efforts to use deep learning to address these challenges and integrate large data sets and computational methods to enable real-time clinical information. Examples are drawn from cardiovascular fluid mechanics, soft-tissue mechanics, and bone biomechanics. The application of deep-learning convolutional neural networks can reduce the time taken to complete image segmentation, and meshing and solution of finite element models, as well as improving the accuracy of inlet and outlet conditions. Such advances are likely to facilitate the adoption of these models to aid in the assessment of the severity of cardiovascular disease and the development of new surgical treatments.

## 1. Introduction

Biomechanics play a central role in normal functions, ranging from the subcellular to the organism level. Forces acting on cells regulate the normal development of the bones, muscles, heart, blood vessels, and other tissues. Soft tissues are in a state of mechanical hemostasis, responding to the normal forces acting on them to establish a narrow range of mechanical properties and maintain normal functioning [1]. Acute or chronic alterations in the biomechanical loads on a tissue or organ induce adaptive responses in the tissue to maintain mechanical homeostasis. Such responses involve gene expression regulating cell growth, differentiation, or extracellular-matrix production. For example, prolonged elevations in blood pressure induce adaptive changes to maintain the overall stress on arteries, causing a decrease in the arterial lumen diameter and an increase in arterial wall thickness [2], and cardiac hypertrophy can arise from weakness in heart muscle after myocardial infarction. Serious injuries can elicit responses that distort mechanical homeostasis, leading to conditions such as osteoarthritis [3] or brain injury [4]. Further, alterations in biomechanical properties due to genetic mutations influence muscular dystrophy [5], Marfan’s syndrome, and hypertrophic cardiac myopathy [6], among others. Considerable insights into these pathological changes have been gained from experimental and computational studies undertaken to characterize normal and pathological biomechanical behavior in tissues.

Biomechanical analyses enhance the understanding of the influence of forces on tissue function, as well as disease initiation and progression, and they serve as tools to diagnose or treat diseases. Imaging techniques provide detailed information on organ geometry, movement, and composition, as well as fluid flows and, thus, they can aid in the formulation of problems, the definition of boundaries and boundary conditions, and the evaluation of results. Several software packages use this patient-specific information on vessel geometry to compute blood flow and fluid stresses acting on the heart chambers, valves, and arteries to diagnose the severity of diseases and guide treatment. This patient-specific information provides the spatial boundaries of the tissue region under study, the boundary and initial conditions, and comparative information to apply computational models of tissue deformation and flow. These models can then accurately calculate quantities describing key mechanical features, such as the strain, stress, pressure, velocity, and flow distribution. Such information can be related to clinical measures of the severity of the disease, such as the fractional flow reserve, which is a measure of the extent to which atherosclerotic disease reduces blood flow in a coronary artery [7]. Parametric studies to examine the effects of the geometry and tissue properties provide further insights into the biomechanical responses of a tissue but are computationally intensive.

To fully understand the ways in which biomechanics influence the disease process, the tissue microstructure needs to be incorporated into biomechanical models, and the subsequent growth and tissue remodeling predicted. The computations for these coupled models are highly time- and resource-intensive, and they are not currently fast enough to provide patient-specific results in real time.

While analytical and computational techniques have been developed to improve various steps in the imaging and computation processes, the methods of machine learning and deep learning can accelerate the overall processes, enabling biomechanics to be an effective tool to aid clinicians in the diagnosis and treatment of diseases. The integration of deep-learning techniques with various steps in the process of analyzing patient-specific biomechanical states in specific organs, a process known as physics-inspired neural networks, offers the opportunity to reduce computation time and improve accuracy by constraining the neural network [8,9], improving the potential for patient-specific applications. The ways in which deep learning facilitates the computational speed and accuracy of the patient-specific simulation of biomechanics and progress in the production of real-time information for clinical decisions are reviewed.

## 2. Computational Biomechanics

Non-invasive imaging methods such as magnetic resonance imaging (MRI), computed tomography, and ultrasound provide critical information about a patient’s health and disease state, as well as geometric and boundary conditions for biomechanical models. These models are based upon conservation relations and specific constitutive equations and fall into three general classes: fluid mechanics, solid mechanics, and gait analysis. To exemplify the various approaches in which deep learning is used, this review examines solid and fluid mechanics models related to soft tissues, with a focus on the cardiovascular system.

For an incompressible fluid, such as blood, the conservation of mass is:(1)∇·v=0
where ***v*** is the velocity vector. The conservation of mass places constraints on the velocity components and the outlet-velocity profile. 

While blood is a non-Newtonian fluid, many computational fluid dynamics (CFD) simulations treat blood as a Newtonian fluid, and this form of the conservation of linear momentum is known as the Navier–Stokes equation.
(2)$$\frac{\partial v}{\partial t}+v\cdot \nabla v=-\nabla p+\rho g+µ\nabla^{2}v$$
where *p* is the pressure, *ρ* is the fluid density, and µ is the fluid viscosity. 

The derivation of Equations (1) and (2) is covered in many textbooks on fluid mechanics and biomechanics [10,11]. Typical boundary conditions are no slip at the vessel wall (i.e., the fluid velocity equals the velocity of the endothelial surfaces of blood vessels) and prescribed pressure and velocity field at the inlet. For patient-specific studies, often, either the inlet and outlet conditions are unknown or limited information is provided on, for instance, the average velocity, flow rate, or specific pressure, and the spatial and temporal variation of the velocity needs to be assumed at these boundaries. 

The suitability of treating blood as a Newtonian fluid has been investigated in a number of studies. When the shear rate (the derivative of velocity with position) exceeds 100 s^−1^, blood behaves as a Newtonian fluid [12], and deviations from Newtonian behavior occur in regions of low fluid velocity and velocity gradients. Deviations from Newtonian behavior affect particle trajectories [13] which may be important in studying monocyte adhesion to inflamed arteries.

The positions of the vessel wall and cardiac muscle change over time due to the pressure field and the mechanical properties of the vessels and heart. Related fluid-structure models range from relations that describe the constitutive relations for the passive behavior of arteries under large strains (see below) to descriptions of arterial-wall remodeling to altered flow conditions and pressure. For most flow simulations, the wall motion is ignored, or a function representing the movement of the vessel wall is imposed, simplifying the solution. 

While conservation and constitutive relations are developed for tissue mechanics, soft tissues can undergo large deformations, even under normal physiological conditions. They exhibit nonlinear elastic or viscoelastic behavior, and are best modeled by nonlinear constitutive equations, which are most easily represented in terms of a scalar strain energy function, ψ. The strain energy function is related to the second Piola–Kirchhoff stress tensor, ***S***, in terms of the Cauchy–Green deformation tensor, **C** [10]:(3)$$S=2\frac{\partial \psi }{\partial C}+pC^{-1}$$
where *p* is the hydrostatic pressure. For *ψ*, a functional form can be assumed based on the tissue mechanical behavior [14] or a microstructural model developed in which the strain energy depends on the stress–strain properties of collagen and the distribution of collagen fibers in the tissue [15]. To solve these problems, one needs the geometry, material properties, and applied loads, as well as other boundary conditions. Obtaining an appropriate constitutive relationship is a key challenge because many soft tissues are anisotropic and the material composition may be heterogeneous, particularly in disease states. The computational methods used to solve these solid or fluid mechanics equations and the associated boundary conditions include the finite element method [16] and the lattice Boltzmann method [17]. The constitutive models are often tested by applying computational models of in vitro experiments featuring biaxial extension [14], in which can the boundary conditions can be imposed by experimental designs.

## 3. Patient-Specific Computational Analysis

The process for modeling patient-specific data is summarized in Figure 1. A variety of imaging methods provide critical information about a patient’s health and disease state, as well as the geometry and boundary conditions for biomechanical models. For CFD studies of blood flow in the heart or blood vessels, advances in dynamic magnetic resonance imaging (MRI) have produced increased detail about vascular-flow fields that can aid in diagnoses. The most advanced of these methods use phase contrast to provide the three-dimensional velocity vector over the entire cardiac cycle (4DFlow) [18]. The velocity or pressure field can then be used to determine the pressure gradients, fluid streamlines, kinetic energy, and wall shear stress, which are important in diagnosing a variety of pathological conditions in the heart and arteries. Errors can be introduced into 4DFlow by acquisition (e.g., spatial and temporal resolution) and processing (e.g., segmentation), as well as the patient (e.g., heart-rate variability, motion) [18]. Such errors can limit the accuracy of the velocity field near the surface of the tissue.

For models of soft tissues, the vessel geometry and microstructure are needed. A variety of atlases contain data on organ structure [19] and cell composition [20]. To examine the microstructure, the collagen-fiber orientation can be determined from second harmonic generation, two-photon microscopy, and other imaging techniques [21]. Image segmentation is a critical process used to obtain the three-dimensional tissue structures of tissues or tissue regions under normal and pathological conditions. The objective is to distinguish the tissue structure from the image background. The geometric information obtained from the segmentation is then used to define the field of analysis through the numerical method. Open-source packages for image segmentation, such as VMTK for tubular structures [22], 3D Slicer www.slicer.org (accessed on 6 September 2023) [23], and Biomedisa [24] can simplify the segmentation process. 

The finite-element-model domain is divided into fine regions with a mesh that accounts for the geometry and the expected gradient of the variables under study. Approximate forms of differential equations are written over these individual mesh components and adjacent mesh elements share common values for the variables. With structured meshes, the element size is fixed, and the geometry makes their application straightforward. However, when variables change rapidly over a given region, unstructured meshes are preferred, since they may reduce the overall number of equations to solve, shortening the computation time, which is beneficial for parallel computing [25]. A particularly useful, but complex, approach involves isogeometric analysis, which is adapted from computer-aided design and creates mesh elements that match the geometry of the tissue domain [26].

Specifying patient-specific inlet and outlet conditions can be challenging. In flow problems, often, only the time-varying flow rate or pressure are known. The inlet flow may be approximated by the Womersley equation for unsteady pulsatile flow in a rigid straight cylindrical vessel [11], but this velocity profile does not capture the complexity of the pulsatile inlet flow field arising from the vessel curvature, short entrance lengths, and the presence of pulse-wave reflections, producing considerable patient–patient variability. For the outlet, the downstream conditions can affect the solution. One approach is to approximate the downstream conditions with a simple one-dimensional model incorporating the resistance and capacitance of the vasculature, known as a Windkessel model [27]. Since the specific flow conditions may vary among individuals and limited downstream information is known, these approximations introduce some errors into the process. Algorithms that optimize the Windkessel model’s parameters from systemic values of pressure and flow are superior to the manual adjustment of parameters for outlet conditions, improving accuracy [28]. An alternative approach is to couple the 3D CFD model with a closed-loop one-dimensional model [29]. Various constraints are needed to ensure that the model is stable and well-conditioned [30]. Various fitting approaches have been developed to obtain the lumped parameter models [31].

Numerical solutions are often tested against known analytical solutions to establish that a particular computational approach is valid. The grid density is tested by determining whether a denser grid yields the same solution within a specified tolerance, which is especially important for fluid–structure interaction models [32]. Once these results are satisfactory, the simulation outputs that are examined include the flow field, streamlines, shear stresses, fluid-residence time, pressure drops, and vorticity in fluid-mechanics simulations, and stresses and strains in solid mechanics models. If microstructural models are included, then the impact of the microstructure features on the stress and strain are evaluated, often requiring additional simulations. Table 1 summarizes some of the patient-specific models and their findings.

While these models can lead to improved diagnostic criteria, a number of limitations need to be addressed. These include the limitations of the imaging technology in the provision of sufficient resolution, the need to validate the model with clinical data, detailed information about loading conditions, and the need for models to incorporate features that depend on age, sex, and race in order to account for population variability [41]. While developed to address orthopedic models, many of the critiques are valid for all biomechanical models. In many tissues, there are numerous unanswered questions about the properties and heterogeneity of the material properties [42]. To improve the computational speed for real-time applications, codes are optimized to run efficiently and parallel computation is used [43]. Key bottlenecks in processing are in mesh generation and computation and in image segmentation. Machine- and deep-learning tools offer efficient methods that can reduce computation times.

## 4. Machine-Learning and Deep-Learning Techniques

Artificial intelligence (AI) is a suite of methods through which computers learn from and make decisions using large quantities of data. The data can take a range of forms, including images, text, sound, or computer simulations. In the simplest applications, computer codes are used to analyze data and extract information. More advanced techniques of machine learning and deep learning are used to improve or accelerate the analysis of medical images or the computation of biomechanical responses (stresses, deformation, or material properties).

Machine learning (ML) programs to recognize patters and involve training programs to represent features of language or text, enabling computers to adapt without human intervention. The program learns the rules governing the features of the data by analyzing large sets of data [44]. In *unsupervised learning,* the software identifies patterns in data using machine learning and some rules without additional human intervention. The data are analyzed by clustering based on similarities, sets of rules relating to variables, or dimensionality reduction, such as in principal components analysis [44]. *Supervised learning* uses labeled data sets to train machine-learning programs using classification algorithms or regression, which mathematically relate input and output data sets [44]. A challenge with machine learning is the need for large sets of high-quality data for training and validation [45]. For patient-specific applications, this may prove challenging. This limitation can be addressed by using finite-element simulations for the training set, but the number of training sets that can be used are limited, in turn, by the computational time [46]. Nonetheless, this approach showed that decision-tree machine-learning algorithms can reduce the computation time of biomechanical models and accurately predict flow in the left ventricle [46].

Neural networks (NN) are training-method algorithms that learn by identifying the relationships among the features in data (Figure 2). These networks consist of a minimum of three layers. The input layer contains information in the data (e.g., information in each pixel of an image), which is transferred to one or more layers, known as the hidden layers, in which features of the data are transformed into a probability. Each node in the network is defined by a weighting factor, bias, and activation function. The activation function provides a nonlinear relationship between the input to the node and its response. Although only one hidden layer is shown in Figure 2, many neural-network programs have multiple layers, each of which has different sets of coefficients, and different transformations in each layer, corresponding to different features in the data. The parameters in each layer are adjusted in response to the training data. The results from the hidden layer(s) are then transferred to the output layer, which transforms the probabilities into the desired outputs. In fluid-flow problems, the inputs are time, t, and position (x, y, z), and the output variables are the velocity components (v_x_, v_y_, and v_z_) and the pressure (p). 

Deep learning (DL) uses multiple hidden layers in neural networks to relate inputs and outputs (Figure 2). Instead of using a rules-based approach, the hidden layers in the deep-learning neural network take advantage of the fact that the data for many types of inputs, such as images, can be transformed into a multi-level structure. For example, the adjoining of pixels with similar intensity may define a border or a specific structure. By training on many images of the same overall structure, the neural network identifies features in another image. The learning process is facilitated by the stochastic gradient method, which utilizes the chain rule of differentiation to rapidly compute the weighting factors and bias [47].

The training set can consist of experimental measurements or computer simulations, and a loss function is established to minimize the mean-square difference between the data and the results of the network. The network is then applied to a test set of data to generalize the learning process [47]. This approach is more accurate and adaptable than machine learning, but it needs a considerable amount of data for training and is computationally intensive. Convolutional neural networks are commonly used because they are easier to train [47], often requiring fewer training sets than machine-learning techniques.

Physics-informed neural networks use the boundary conditions, material property values, and/or partial differential equations governing the biomechanical problem to place constraints on neural networks. This addresses two limitations in the use of deep learning to solve biomechanical models: one, the large quantity of data needed for training and two, deep learning can identify solutions within the range of the data provided for evaluation but cannot extrapolate and predict results for other conditions or geometries not included in the training set. 

The overall approach to integrate model physics into the deep-learning algorithm is as follows [9] (Figure 3):Step 1.Construct a neural network that predicts a solution for a key variable (e.g., velocity, stress) from the inputs using the parameters of the neural network.Step 2.Specify the two training sets for the equation and boundary/initial conditions. These data define the problem under study.Step 3.Specify a loss function between the neural-network output and both the PDE and the boundary-condition residuals.Step 4.Train the neural network to find the best parameters for the network that minimize the loss function. The stochastic gradient method provides a rapid algorithm to obtain the neural network’s parameters [48].


The loss function constrains the parameter space over which the neural network operates to compute parameters.

An important feature of neural networks is that they can perform differentiation automatically, eliminating the need to perform operations on a finite-element grid, accelerating computation [8] and facilitating applications for grid optimization.

## 5. Machine-Learning and Deep-Learning Applications to Computational Biomechanics

Machine- and deep-learning methods have been applied to many of the steps in Figure 1 to develop and solve patient-specific biomechanical models, including image segmentation, image analysis to extract specific features, mesh generation, and the solution of biomechanics equations. Examples from each of these specific topic areas are considered. A number of datasets are available. For blood vessels, www.vascularmodel.com (accessed on 6 September 2023) contains images, anatomic models, and finite-element models for the simulation of over 250 patient-specific geometries. These datasets interface with the Simvascular modeling platform [49]. The Cardiac Atlas (https://www.cardiacatlas.org/ (accessed on 6 September 2023)) contains cardiac datasets that can be accessed through data-use agreements, with the institution providing the data [19].

**Image Segmentation** is one of the most challenging issues in model development. Errors can arise as a result of incomplete discrimination from the background due to noise, poor contrast, artifacts due to the presence of other organs, and complex geometry. A range of computer-vision approaches have been used to perform many of the tasks to improve upon manual operations, including thresholding, pattern recognition, optimization methods, and combinations of these techniques [50]. While these methods work well on high-contrast images, their performance is limited with lower-quality images and with the large data contents in modern imaging methods. A variety of current and emerging deep-learning approaches to image segmentation were recently reviewed [51]. In addition, the current state of medical segmentation algorithms, methods to evaluate segmentation approaches, and applications in the chest, abdomen, and specific organs was reviewed recently [52].

The 2D and 3D U-net packages [53,54] use very few training images by performing elastic deformations on the images, enabling convolutional neural networks to learn about the invariance of deformation. Algorithms have been developed to resolve issues related to the touching of objects, resulting in more precise segmentation. By using the 2D and 3D forms of U-net and a series of rule-based parameters, nnU-net optimizes choices for reconstructed images [55]. 

The use of a loss function that focuses on a region near the vessel wall reduces problems with segmentation using balanced loss functions [56]. The deep-learning approach increased the automation of the segmentation process, increasing the overall throughput, and performed slightly better than U-net. In a subsequent study, a convolutional neural network was developed to extract a vessel-lumen boundary from 2D CT and MR images. This approach showed improved accuracy in determining vessel dimensions without other processing steps.

Iyer et al. [57] developed AngioNet, a two-step convolutional neural network procedure for the automatic segmentation of blood vessels. First, an angiographic processing network was applied to improve the contrast and the sharpness of the vessel boundaries without the a priori selection of a particular pre-processing filter. This output was then used by Deeplabv3+ [58] to create a segmented image. The accuracy was 0.18 ± 0.24 mm, which was at the level of 1–2 pixels. An advantage of this approach is the overlapping structures (e.g., bones, catheters), eliminating the need for manual actions to improve the image. However, the software can overpredict vessel boundaries in severe stenosis.

**Image Analysis for Feature Characterization.** In addition to providing structural information, imaging methods can provide details on blood flow, flow in other tissues, and detailed structural information. 

While 4DFlow provides detailed information about these flow fields, the images can be noisy, with inaccurate velocity values near vessel walls, creating challenges in the estimation of quantities based on derivatives of the velocity, such as the vorticity and wall shear stress. A variety of regression and filtering techniques have been used to address these limitations (e.g., [59]). Using synthetic MRI data generated by CFD simulations in the thoracic aorta and modified to be consistent with the types of data generated by MRI, a deep learning model reduced noise and improved spatial resolution, although the assessment was qualitative [60]. This neural network, which was trained on aortic-flow images, significantly reduced noise and enabled the calculation of the fluid vorticity in the right and left ventricles of adults who received surgery to address tetralogy of Fallot [61]. The results indicated that the changes in the right- and left-ventricle shape in these patients were affected by the vorticity.

Alternatively, computational fluid dynamics can be used to assess the consistency of 4DFlow. To address errors in the method, Rutkowsi et al. [62] used convoluted neural networks trained on computational fluid dynamic simulations of vessel geometry to enhance the MRI velocity images of five cerebral aneurysm vessels. From these five image data sets, twenty additional models were manually modified, and six sets of unique inflow conditions were used with each geometry for a total 180 CFD simulations. The simulations provided the training set for the convolutional neural network (CNN), which was then evaluated against time-averaged 4DFlow MRI data from 20 patients. The CNN-enhanced velocity images had lower noise and higher apparent spatial resolution than the raw velocity images and greater vessel-boundary delineation. This improved resolution led to corrections in the fluid streamlines (Figure 4). A limitation of this approach is the computing costs for training sets, since separate training sets are required for each type of vessel.

By teaching neural networks to learn the relationship between fine and coarse grain flow fields through the choice of the neural network and a loss function with directional sensitivity, the super-resolution of the flow field can be achieved from 4DFlow data using smaller training sets and less computation time [63].

**Solution of Conservation and Constitutive Relations.** Given the power of deep learning, some investigators have examined the feasibility and usefulness of solving conservation relations and constitutive equations with neural networks using imaging data to train neural networks. Neural networks can be trained with experimental or simulated data related to force and deformation to predict property data, as well as to simulate fluid flow, tissue deformation [64], or force deformation [43], leading to a significant decrease in computation time. This approach is most beneficial when estimating the material properties for tissue constitutive relations. The use of finite-element simulations of the governing conservation and constitutive relations as training data can reduce the amount of patient data needed. Such simulations need to be validated against previous patient data to ensure their reliability. 

To establish the feasibility of identifying key hemodynamic parameters describing a disease state, Feiger et al. [65] used a single-patient aortic-coaction geometry and generated training and test sets from 50 simulations by varying the viscosity and flow rate. Furthermore, they used a single hidden-layer neural network to predict the pressure drop. The agreement between he neural-network model and the lattice Boltzmann model was excellent. The model also predicted the pressure drop for levels of stenosis other than those present in the single patient sample. The prediction of other hemodynamic parameters was less accurate.

A five-layer convolutional neural network was used on second-harmonic-generation images of heart-valve tissue to predict stress–strain curves from equibiaxial testing on 48 test samples [66]. The model predicted the overall shapes of the stress–strain curves, but with an offset. The overall accuracy was 84%.

The deformation of a porcine tricuspid valve was predicted from digital image correlation (DIC) using implicit Fourier neural operators [67], which incorporates long-range features by using an integral operator. The DIC was used to analyze the valve displacement during biaxial testing. The neural-network model exhibited less error than a finite-element structural model but required at least 10,000 training samples. However, the neural-network model alone was not particularly accurate with conditions other than those used for the training set.

**Mesh Generation** is critical for obtaining accurate solutions from finite-element models. The packages available for the generation of meshes form solid mechanics and fluid-mechanics models [68], but their usefulness is limited when the model is multi-scale, due to either the nature of the mechanics or the incorporation of the tissue microstructure. In both cases, the fine structure is much smaller than the domain of the computational model, placing significant constraints on the computational time and cost [69]. Further, the integration of microstructural models into finite-element models of soft tissues influences the mesh structure, since the collagen fibers have specific regional orientations [70]. While deep-learning approaches are appealing, training sets may be limited or non-existent for microstructures. 

To analyze the effect of the microstructure on the fatigue of bioprosthetic heart valves, Zhang et al. [70] developed a neural-network representation of the first term on the right-hand side of Equation (3) for the strain-energy function, $2\frac{\partial \psi }{\partial C}$. The training set was then developed by assuming a functional form for the fiber-orientation distribution and then computing the strain-energy function. This approach enabled a more rapid development of the mesh for parametric studies (Figure 5). This model used an isogeometric analysis (IGA), in which the mesh elements match the exact geometry using non-uniform rational B-splines [16]. This focused neural-network approach is as fast as deep-learning models, which replace the entire set of conservation and constitutive relations, and retains the ability to study parametric sensitivity.

**Physics-Informed Neural Networks (PINN).** The limitations of using deep-learning neural networks to solve conservation relations are the large amount of training data needed and the lack of a guaranteed physically consistent solution. To take advantage of the computational speed gained from the neural networks and to produce a reasonable solution, the loss function incorporates the boundary conditions and conservation relationship (Figure 3). Nondimensionalization scales the variables in a physically consistent way, making the neural network robust and improving training [71]. Downstream effects can be optimized by incorporating the general Windkessel model in the loss function, allowing the neural network to find the best values for peripheral resistance and capacitance [71]. Since these constants are difficult to obtain by other methods, this approach can improve the overall solution accuracy. This approach was used to compute velocity and velocity gradients near the arterial surface from sparse data sets and incomplete boundary conditions for several idealized cases, including the blood flow in an aneurysm [72].

The PINN can be used to produce the super-resolution of the flow field from low-resolution and noisy training sets, even when the inlet boundary condition is not known [73]. 4DFlow data can be used directly without further refinement. This approach enables the exploration of a large parameter space much more rapidly than would occur with CFD solutions for the various parameter values. 

While several microstructural models of arteries have been developed, estimation of the material properties that are valid over the entire range of the stress–strain curve has proved challenging. To address this limitation, a hybrid model was developed, in which a deep neural network was used to extract the material properties from second-harmonic-generation images [74]. The parameters were then used to determine the Cauchy stress from the strain-energy function in the microstructural model. The loss function is the mean-square error between the stresses computed in the experiment using this hybrid physics-informed model. The model used a smaller training set than that involved in the direct use of the neural network to predict stress, and it gave superior fits for the stress–strain data than the direct fitting of the microstructural model. A limitation of this approach is that the hybrid model gave poor fits for the data outside the range of the training set. 

To establish the microstructural parameters affecting the mechanical behavior of the mouse aorta, Linka et al. [75] used constitutive artificial neural networks, which receive as inputs strain invariants and information about the microstructures of tissues. Their output is the strain-energy-density function and the associated stress. The constitutive artificial neural networks incorporate the general relationships between the stress and the strain (e.g., Equation 3 and other relationships), but do not specify the forms of the constitutive relationships. Generally, this approach yielded very good fits, with r^2^ above 0.9. The model was also used to identify the microstructural parameters that most significantly affected the fits.

Taking advantage of the ability of neural networks to perform automatic differentiation, neural networks were used in a cardiac model to compute the displacement field from pressure and active contraction inputs [76]. Both a simplified model and a complete mechanical model were considered for the training. The biomechanical model and the boundary conditions were incorporated by minimizing the residual of the force vector between the neural network and the simulation. The simplified model worked as well as the complete model, but required less data for training and was faster. After training, the model was able to predict the twisting motion of the cardiac muscle in agreement with experimental results.

One of the most significant challenges in computational biomechanics is the integration of multiscale models. A proof-of-principle study showed that PINN can be used for the multiscale modeling of thrombus formation [77]. The approach incorporated the Navier–Stokes equation and a model that incorporates the steps in thrombus formation (platelet transport, activation, and aggregation) and deformation by fluid flow.

## 6. Conclusions

Rapid advances have occurred in the incorporation of machine and deep learning into the various aspects of computational biomechanics. These have led to improvements in the quality of imaging data, enabling the more efficient segmentation of images and the accurate computation of shear stresses and vorticity. Deep-learning algorithms can accelerate the mesh-generation process, decreasing computation times. In parametric sensitivity studies, deep-learning neural networks can replace finite-element solutions, reducing overall computation times. Novel approaches have been developed to reduce the number of training sets. These techniques can also aid in identifying key microstructural parameters and suitable microstructural models. 

While these gains are impressive and can be used to analyze patient-specific data, their full introduction into clinical practice for real-time diagnosis is becoming achievable. To this end, it is necessary to improve the ability to further reduce computational times, establish a wider set of test cases, and predict accurate results outside of training regimes. As microstructural information is incorporated into these models, the uniqueness of the compositions of various output measures needs to be considered. As has been shown with numerous artificial intelligence approaches, bias can enter into the learning process, either explicitly or implicitly. To address this, training sets need to encompass variations due to age, sex, and race to ensure that the predictions truly reflect what data represent about specific individuals’ health. 

## Figures and Tables

**Figure 1 bioengineering-10-01066-f001:**
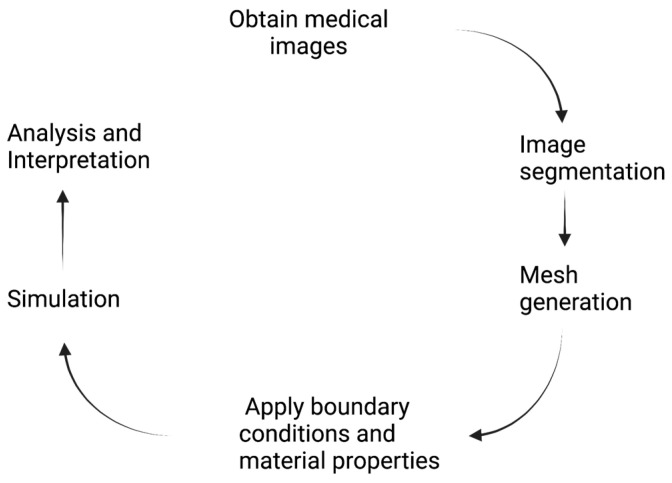
Schematic of the steps in the simulation of patient-specific biomechanics.

**Figure 2 bioengineering-10-01066-f002:**
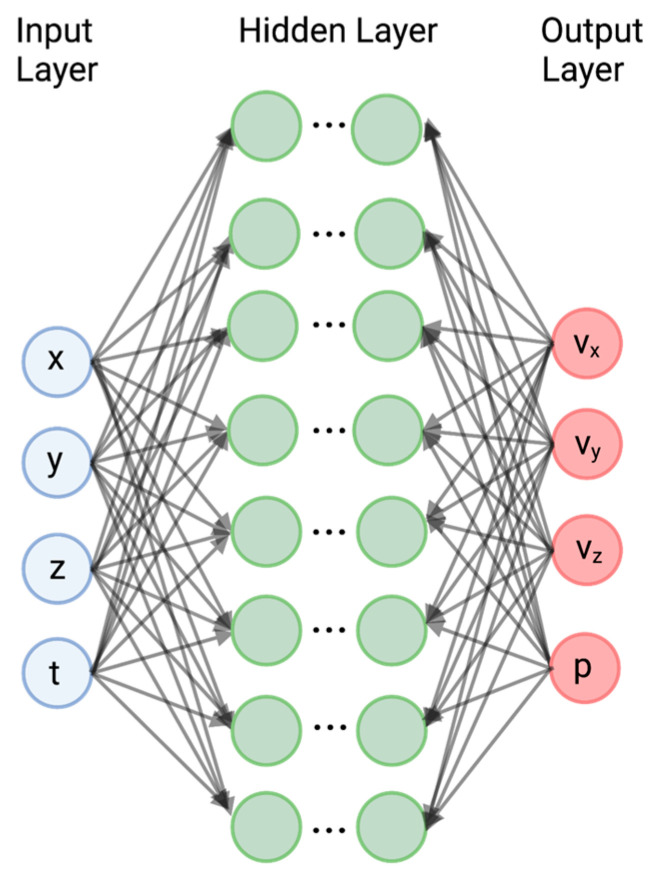
Schematic of neural networks for prediction of fluid velocity and pressure from training sets consisting of experimental measurements and computer simulations. The green circles represent the different layers of the neural network. The red circles represent the derived parameters (v_x_, vy, v_z_, and p) from the neural network.

**Figure 3 bioengineering-10-01066-f003:**
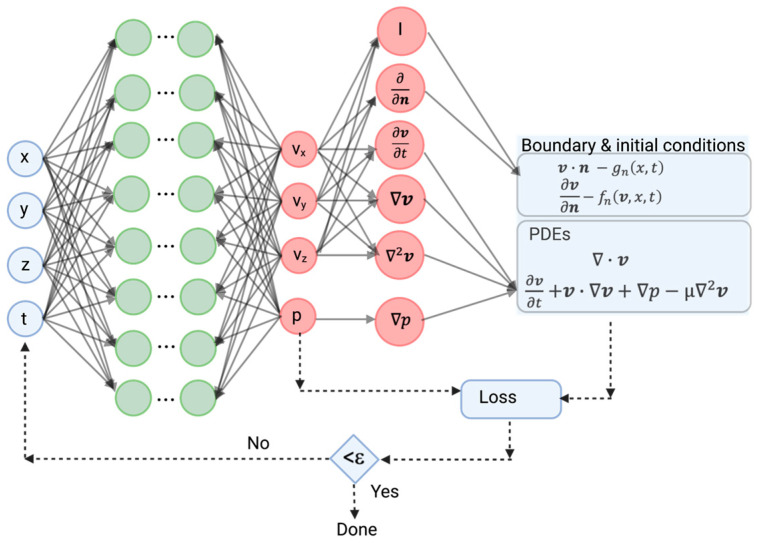
Schematic of physics-informed neural networks (PINN) applied to a solution of the Navier–Stokes equation and boundary conditions. I represents the identity matrix, **n** is the normal unit surface facing into the fluid. The green circles represent the different layers of the neural network. The red circles represent the different operations performed on the derived parameters (v_x_, vy, v_z_, and p) from the neural network.

**Figure 4 bioengineering-10-01066-f004:**
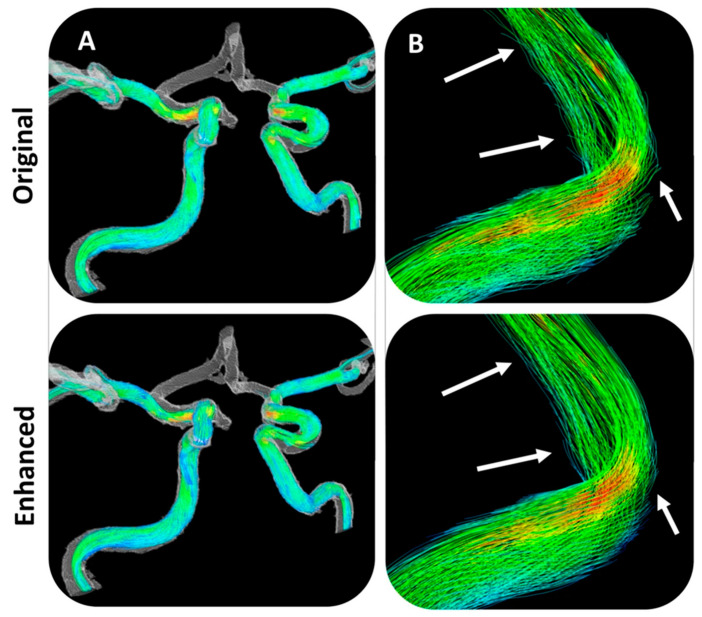
(**A**) Streamlines at low resolution through the right and left internal and middle carotid artery are similar between original and enhanced flow fields. A higher-magnification view of the internal carotid artery bend in (**B**) shows that streamlines from the original flow field tend to point and terminate outside the vessel wall, while streamlines in the enhanced flow field retain normal flow paths. From [62] and published based on a CC-BY Creative Commons license from Scientific Reports.

**Figure 5 bioengineering-10-01066-f005:**
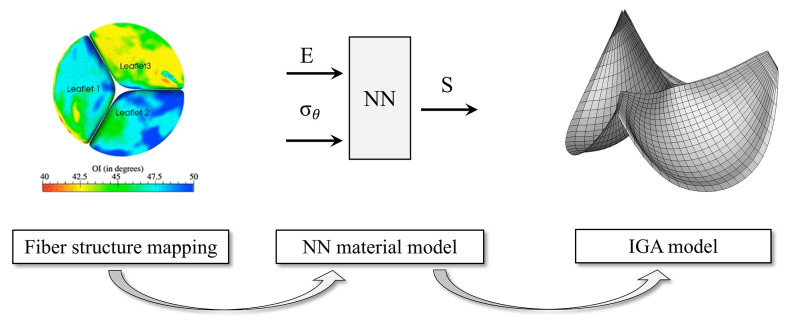
Computational process using neural networks to map the fiber structure to mesh for isogeometric analysis (IGA) model of bioprosthetic heart valves. E—Green–Lagrange strain tensor; σ_θ_—standard deviation of preferred fiber direction; S—second Piola–Kirchhoff stress tensor (Equation (3)). Reprinted from [70], with permission.

**Table 1 bioengineering-10-01066-t001:** Selected patient-specific applications of computational biomechanics.

Topic	Reference	Key Results
**Orthopedic Biomechanics**		
Design of stem for total hip arthroplasty	[33]	Identified novel design that produced strains comparable to those present before surgery.
Oral and maxillofacial surgery	[34]	Survey of various finite-element models in trauma and reconstructive surgery and implant design.
Modeling of bone	[35]	Overview of processes to model deformations, and implant interactions
**Cardiovascular Biomechanics**		
Mitral valve repair	[36,37]	Model developed from 3D transesophageal echocardiography; workflow for steps in Figure 1.
Abdominal aortic aneurysms	[38]	Wall shear stress is a critical factor affecting rupture and can be predicted with four geometric parameters, which can be measured.
Single functional ventricles	[39]	Fluid-structure-interaction model indicates that a common surgical procedure can be modeled assuming rigid vessels.
Coronary-artery fractional -flow reserve	[40]	Identification of minimal number of patient variables to estimate fractional flow reserve

## Data Availability

Not Applicable, this was a review.

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
