# Peer review of "The Potential of Deep Learning to Advance Clinical Applications of Computational Biomechanics"

_bioengineering, 2023, doi:10.3390/bioengineering10091066_

Round 1

Reviewer 1 Report

A great review about current challenges in using computational data for ML/DL training. 

Boundary conditions are key part of computational studies but these conditions are not mostly patient-specific. More discussion about potential ways of dealing with boundary conditions may be helpful.

Author Response

RESPONSE: Thank you for this helpful suggestion.  The following text was added to expand on the details about boundary conditions.

1. Lines 148-150. Mention how boundary conditions can be simplified for in vitro testing on constitutive relations for soft tissue.

2. Lines 191-207. We expanded this section, providing additional details about the boundary conditions and included additional references.

Reviewer 2 Report

The author shows a review of the state-of-the-art methods applied in computational biomechanics based on deep learning techniques. The paper is well written and structured, although there are some points that should be reconsidered.

Specific comments:

1 Introduction

Page 1 Lines 24-25. I guess you mean “mechanical homeostasis” here, “hemostasis” is a different concept.

2 Computational Biomechanics

Page 2 Line 75. Why gait analysis? The mentioned non-invasive imaging methods are note able to provide results regarding the gait analysis.

Page 3 Line 103. “. are best modeled…” Please check this sentence.  

Eq. (3). Please note that the strain energy density function in this expression corresponds to the isochoric part and that the jacobian is missing in the second term multiplying the hydrostatic pressure. You could find the proper expression in reference [13]

3 Patient-Specific Computational Analysis

Page 4 Lines 149-150. “However, when variables change rapidly over some region unstructured meshes are preferred since they may reduce the overall number of equations to solve, reducing computation time, which is beneficial for parallel computing.” This is just the opposite, when the gradient of the variable is large over one region, it is necessary to refine the mesh, i.e., reduce the number of elements. In this case the computation time increase. Please rewrite this sentence.

Page 5 Lines 196-197. “Key bottlenecks in processing are in mesh generation and computation, image segmentation.” Please rewrite.

4 Machine Learning and Deep Learning Techniques

Page 5 Lines 207-209 “Machine learning (ML) programs to recognize patters and involves training pro-207 grams to represent features of language or text enabling the computer to adapt without 208 human intervention.” This sentence is hard to follow.

Page 5 Line 222. Please reconsider the definition of NNs, they are not a training method. I also recommend modifying the following lines, the simplest neural network could be just a single layer where the inputs are connected directly to the outputs. 

5. Machine Learning and Deep Learning Applications to Computational Biomechanics

Page 10 Line 390. Remove the repeated “in which”.

Author Response

Reviewer #2

Comments and Suggestions for Authors

Author presented a nice and well written review study on the integration of machine learning, artifical intelligence, neural network, and physics informed neural networks in computational biomechanics.

Just a minor recommendation for your consideration, in the Abstract, could author add what might be the pratical implications for clinical medical imaging and biomechanics from this review study?

RESPONSE:  Thank you for the suggestion.  We have added the following sentences to the Abstract “Application of deep learning convolutional neural networks can accelerate the time to complete image segmentation as well as meshing and solution of finite element models and improve the accuracy of inlet and outlet conditions.  Such advances are likely to facilitate the adoption of these models to aid in assessment of cardiovascular disease severity and development of new surgical treatments.”

Reviewer 3 Report

Author presented a nice and well written review study on the integration of machine learning, artifical intelligence, neural network, and physics informed neural networks in computational biomechanics.

Just a minor recommendation for your consideration, in the Abstract, could author add what might be the pratical implications for clinical medical imaging and biomechanics from this review study?

Author Response

RESPONSE:  Thank you for the suggestion.  We have added the following sentences to the Abstract “Application of deep learning convolutional neural networks can accelerate the time to complete image segmentation as well as meshing and solution of finite element models and improve the accuracy of inlet and outlet conditions.  Such advances are likely to facilitate the adoption of these models to aid in assessment of cardiovascular disease severity and development of new surgical treatments.”

Reviewer 4 Report

1) The English language is very weak

2) The need for the review is not clearly mentioned in the abstract

3) The entire article seems to be just a summary instead of the critical review on the existing works. The drawbacks of the existing methods must be clearly pointed out.

4) Tables, bar charts, etc.. can be used to represent the information instead of the plain theory. It makes the article less interesting.

5) More information on the available datasets can be given.

6) Already known information is repeated again in most of the instances.

7) The number of reference articles are too less for a SCIE article

Very hard to understand

Author Response

1) The English language is very weak

RESPONSE: While specific instances were not cited, more specificity and clarity was added throughout the text and can be assessed by examining the version with the tracked changes.

2) The need for the review is not clearly mentioned in the abstract

RESPONSE: The motivation for the article was stated on lines 14-16 of the Abstract “This review summarizes current efforts to use deep learning to address these challenges and integrate the large data sets and computational methods to enable real-time clinical information.”  Further, the last sentence of the introduction (lines 72-74) was revised to the following” The ways in which deep learning facilitates the computational speed and accuracy of patient-specific simulation of biomechanics and progress to produce real time information for clinical decisions are reviewed.”

3) The entire article seems to be just a summary instead of the critical review on the existing works. The drawbacks of the existing methods must be clearly pointed out.

RESPONSE:  Limitations or drawbacks of various approaches are noted in the following locations in the text.

  1. Lines 102-106. Limitations of assuming Newtonian behavior of blood and applying the Navier-Stokes equation are noted.
  2. Lines 143-145. Challenges to developing constitutive relations for soft tissue are noted.
  3. Lines 166-169. Errors in 4DFlow imaging are mentioned. Later in the article, application of deep learning is described to address this problem.
  4. Lines 200-210 Summarize limitations of current CFD modeling.
  5. Lines 276-280. Highlight limitations of direct application of deep learning to the solution of computational models.
  6. Qualifying statements were made about applications of deep learning on lines 399, 408 and 409, 417-419.

4) Tables, bar charts, etc.. can be used to represent the information instead of the plain theory. It makes the article less interesting.

RESPONSE: A graphical abstract was added.  Table 1replaced the text on patient-specific applications.

5) More information on the available datasets can be given.

RESPONSE: We added additional information on datasets on lines 306-311.

6) Already known information is repeated again in most of the instances.

RESPONSE:  Repetition was minimized, but several topics were raised in several sections (e.g. boundary conditions), but addressed within the context of the section title.

7) The number of reference articles are too less for a SCIE article.

RESPONSE: Additional references were added, bring the total to 77 from 65.  A number of references are reviews that provide more detailed exposure to the topics noted.

Reviewer 5 Report

I have had the pleasure of reviewing your manuscript on the integration of deep learning in computational biomechanics for clinical applications. I must commend you on your adept handling of such a multifaceted topic. Your article strikes an admirable balance between conciseness and comprehensiveness, offering readers a clear and insightful overview of the current landscape, while also highlighting the immense potential in this field. I genuinely appreciate your contribution to the field and look forward to seeing more of your work in the future.

Author Response

Response: Thank you very much for the positive feedback.

Round 2

Reviewer 4 Report

It can be accepted now

Minor editing can be done